# The Potential of Purple Waxy Corn Cob (*Zea mays L*.) Extract Loaded-Sericin Hydrogel for Anti-Hyperpigmentation, UV Protection and Anti-Aging Properties as Topical Product Applications

**DOI:** 10.3390/ph16010035

**Published:** 2022-12-27

**Authors:** Nattawadee Kanpipit, Natsajee Nualkaew, Suthasinee Thapphasaraphong

**Affiliations:** 1Biomedical Science Program, Graduate School, Khon Kaen University, Khon Kaen 40002, Thailand; 2Department of Pharmacognosy and Toxicology, Faculty of Pharmaceutical Sciences, Khon Kaen University, Khon Kaen 40002, Thailand; 3Department of Pharmaceutical Chemistry, Faculty of Pharmaceutical Sciences, Khon Kaen University, Khon Kaen 40002, Thailand

**Keywords:** sericin hydrogel, anthocyanins, UV protection, collagen stimulating activity, anti-melanogenesis, anti-aging

## Abstract

Sericin-hydrogel formulations incorporating purple waxy corn (*Zea mays* L.) cob extract (PWCC) were developed as potential topical skin cosmetic products. Sericin has wound healing properties, protects against ultraviolet (UV) radiation, and exhibits anti-inflammatory, anti-oxidation, and anti-tyrosinase activities. PWCC is a rich source of anthocyanins with antioxidants, UV protective, anti-inflammatory, and collagen-enhancing activities. Six hydrogel formulations (S1–S6) were investigated for anti-melanogenesis on the B16F10 melanoma cell line and UV-protection on human keratinocytes (HaCaT) and anti-aging activities on normal human dermal fibroblasts (NHDFs). The results showed that the hydrogel formulations enhanced the anthocyanin permeation through the skin. The S4 formulation indicated the highest inhibition of tyrosinase activity and reduced the melanin pigment, increased the cell viability of the UV-induced HaCaT cells, the inhibition of collagenase and elastase, and increased the collagen type I production without cytotoxicity. Therefore, the PWCC loaded-sericin hydrogels show a high potential as a novel anti-hyperpigmentation, UV protection, and anti-aging products for topical applications.

## 1. Introduction

Purple waxy corn (*Zea mays* L.) is a natural resource, rich in anthocyanins that have antioxidant properties. Extracts from each part of the purple waxy corn were investigated for its phenolic and anthocyanin contents and its antioxidant activity, and the cob extracts had the highest anthocyanin content per unit of area [1]. In addition, the highest anthocyanin was obtained from the purple waxy corn cob of the KND strains developed by the Plant Breeding Research Center for Sustainable Agriculture, Khon Kaen University, Thailand [2]. In addition, several by-products of the purple waxy corn extracted from silk and cob were determined and found to be a major anthocyanin, such as cyanidin-3-glucoside (C3G), peonidin-3-glucoside (Pn3G), and pelargonidin-3-glucoside (Pg3G). Both extracts also stimulate the collagen production in fibroblasts [3].

A topically applied purple corn extract was able to inhibit the aging process by reducing the matrix metalloproteinase 1 (MMP-1) levels and increasing the collagen in the skin of rats exposed to UV-B radiation [4]. Furthermore, the purple corn extract provided protection against the ultraviolet B induced cell damage [5] and inflammation by suppressing the NF-κB nuclear translocation and decreasing the proinflammatory mediators, such as iNOS, and the COX-2 levels in human keratinocyte (HaCaT) cells [6]. Finally, the purple waxy corn extracts were found to increase the collagen levels in normal human skin fibroblasts [3].

Sericin is a soluble protein sourced from silkworm cocoons (*Bombyx mori*) with a molecular weight range of 20–400 KDa. Sericin is a biodegradable and biocompatible material suitable for medical and pharmaceutical applications [7]. Sericin is widely used in topical products designed for cosmetic applications, due to its elastase, hyaluronidase inhibition, and UV-radiation protection [8]. Sericin extracted using a high temperature and high-pressure degumming technique was found to accelerate the cell ed from the proliferation and it was found to promote the collagen production [9] and inhibit tyrosinase [10] in the L929 mouse fibroblast cell line. Furthermore, sericin reduced the UVB-induced oxidative damage and hyperpigmentation via the melanogenesis pathway in the B16F10 mouse melanoma cell line [11].

Hydrogels are three-dimensional networks of hydrophilic polymer chains capable of holding large amounts of water that can swell, depending on the environmental conditions, such as the pH, temperature, and ionic strength [12]. Natural polymers that are commonly used in hydrogels, control and improve the active ingredient delivery to target sites by allowing the diffusion through the mesh or pores of the polymeric network without side effects [13]. In addition, the diclofenac hydrogel was also reported to enhance the diclofenac skin permeation and absorption [14]. A previous report stated that an anthocyanin-loaded hydrogel exhibited the enhanced stability and prolonged anthocyanin release, as well as the downregulation of the iNOS, IL-6, IL-1β, and TNF-α pro-inflammatory mediators. These combined anti-inflammatory and wound-healing properties suggest the synergistic effects of anthocyanins and hydrogels [13].

Melanin is a pigment which is synthesized and deposited in melanosomes, which are produced by melanocytes and is transferred to the neighboring keratinocytes throughout the physiological process called melanogenesis [15]. The melanogenesis process occurs in the stratum basale in the epidermis layer of the skin. The role of melanin is to provide protection for the skin against UV radiation and reactive oxygen species (ROS) damage. However, UV can induce the overproduction of melanin. The accumulation of melanin can cause hyperpigmentation and provides evidence of skin aging in the stratum basale. Melanin synthesis in melanocytes is activated by tyrosinase, which catalyzes the oxidation of L-tyrosine to L-3,4-dihydroxyphenylalanine (L-DOPA), which is a rate-limiting step in melanogenesis [16]. L-DOPA is oxidized to form DOPAquinone, which further reacts with cysteine or glutathione to produce cysteinyl DOPA and the benzothiazine derivatives of pheomelanin. The two major forms of melanin, black/brown eumelanin and yellow/red pheomelanin are both initially catalyzed by tyrosinase [17]. Therefore, the inhibitors of melanogenesis via the tyrosinase inhibition are important natural compound targets for the development of skin-whitening agents. There are several natural compounds that inhibit the melanin production by inhibiting tyrosinase via the suppression of MITF or by activating the MAPK/ERK pathway that are non-toxic and exhibit no-side effects [18].

The anthocyanin extract from *Hibiscus syriacus* L. was found to promote the anti-melanogenic activity by activating the ERK signaling pathway [19]. A liposome-encapsulated anthocyanin delivery system inhibited the melanin synthesis through the inhibition of tyrosinase and the suppression of the protein expression in the melanogenesis pathway [20].

UVB irradiation induces the production of the reactive oxygen species (ROS) in keratinocytes, which affects the skin and accelerates the aging process by damaging the cell macromolecules. UV irradiation has been shown to upregulate the MMP genes (MMP-1 and MMP-2) in HaCaT cells. The production of MMP-1 by keratinocytes induces the fibroblasts to secrete inflammatory cytokines, such as IL-1β and IL-6. Therefore, inhibiting the UV-induced upregulation of MMPs in keratinocytes is a target to prevent the fibroblast photoaging. Furthermore, the strong antioxidant properties of the cuatrec extract from *Licania macrocarpa* have been reported to improve the moisturizing conditions, diminish the hyperpigmentation, and protect the skin cells from the effects of UV irradiation [21]. Natural substances from medicinal plants, such as polyphenols, tocopherols, carotenoids, ascorbic acid, macromolecules (including polysaccharides and peptides), and essential oils could promote skin health and protect against various harmful factors, including ultraviolet radiation and free radicals [22].

In this study, we evaluated the anti-melanogenesis and anti-aging properties of the purple waxy corn cob extract incorporated into a sericin hydrogel in human keratinocytes, the B16F10 murine melanoma cell line, and normal human dermal fibroblast. We reported the potential of hydrogels as anti-aging and anti-hyperpigmentation topical cosmeceutical products.

## 2. Results and Discussion

### 2.1. Determination of the C3G, Pn3G and Pg3G Contents by HPLC-MS/MS

The purple waxy corn cob extracted by 50% ethanol was identified and determined by HPLC-MS/MS, as particular anthocyanins, such as C3G, Pn3G, and Pg3G, which were detected in the molecular ions ([M + H]+) at m/z 499, 463, and 433, respectively [3]. The molecular structures of the anthocyanin derivatives (C3G, Pn3G and Pg3G) are shown in Appendix A. The HPLC-MS/MS chromatograms of C3G, Pn3G, and Pg3G are presented in Appendix A, with the retention time of 2.77–2.78 min. The amount of C3G, Pn3G, and Pg3G was 2.42 ± 0.03, 0.99 ± 0.01, and 0.68 ± 0.02 mg/g dried weight, respectively. Therefore, C3G was observed as the highest content in the purple waxy corn cob extract.

### 2.2. Physical Properties of Sericin-Anthocyanin Hydrogels

The extraction and characterization of sericin from silkworm cocoons and anthocyanin from the purple waxy corn cob, was reported previously [12]. The hydrogel formulations (S1–S6) were prepared with a buffer pH 6.5, since anthocyanins are more stable under acidic conditions, and this resulted in final pH values between 6.45 and 6.71. The pH values of the formulations were comparable with the pH of human skin (pH 5.5–6.5). The viscosity of the formulations with alginate 0.2% *w*/*v* were 37.51 ± 0.01 Pa.S (S2), 35.50 ± 0.03 Pa.S (S4), and 34.49 ± 0.05 Pa.S (S6), which were slightly higher than their equivalent formulations without alginate, which had the viscosity of 35.52 ± 0.00 Pa.S (S1), 35.39 ± 0.02 Pa.S (S3), and 34.27 ± 0.04 Pa.S (S5). Furthermore, the viscosity decreased while the concentration of PWCCS increased (Table 1). The total anthocyanin contents of S1–S6 were determined by the pH-differential method and were obtained in the range of 39.21–107.61 mg C3GE/L (Table 1). The optimum hydrogel formulation (S6) contained 0.2% of alginate, which provided a suitable viscosity, sufficient swelling capacity, and the controlled release of anthocyanin [12].

#### 2.2.1. Swelling Capacity

The swelling capacity of a hydrogel is an important physical property related to the controlled release of anthocyanin from the formulation. The swelling capacity (% Swelling) over 48 h of each hydrogel formulation containing alginate (S2, S4, S6), was higher than its respective formulation without alginate (S1, S3, S5) (Figure 1). Swelling allows for the more rapid diffusion of anthocyanins from the hydrogel since the diffusion release mechanism occurs at the outer surface of the polymer, rather than the core porous structure [13]. Previous research reported that the kinetics release of hydrogels was achieved using the Korsmeyer–Peppas model, due to the electrostatic and H-bonding interactions between alginate, anthocyanin, and sericin in the hydrogel, which was confirmed by FT-IR spectroscopy. The release of anthocyanins was by rapid diffusion at the outer surface of the polymers in the hydrogel for up to 4 h, followed by the gradual swelling until 48 h, due to the cross-linking interactions between anthocyanins and the hydrophobic polymers [13].

#### 2.2.2. Skin Permeation

The S1–S6 formulations were evaluated for their ability to penetrate through the skin using a porcine skin model in a Franz diffusion cell. The permeation profile of all formulations (S1–S6) and the 0.15% and 0.5% anthocyanin solutions (PWCCS0.15 and PWCCS0.5) within 24 h are shown in Figure 2. The cumulative amount of permeation of the anthocyanin-loaded sericin hydrogels (S3–S6) showed higher results than the same concentration of anthocyanins in the solution form (PWCCS0.15 and PWCCS0.5). The highest cumulative anthocyanin permeation content in 24 h was from S6. The S5 and S6 formulations containing 0.5% PWCCS showed improved anthocyanin permeation, compared to the equivalent S3 and S4 formulations containing 0.15% PWCCS. This suggested that the higher anthocyanin content enhanced the permeability of anthocyanins.

According to a previous report, the incorporation of sericin in the hydrogel formulations improved the physical properties of the hydrogel by increasing the viscosity and extending the release time and improving the stability of anthocyanins [12]. The hydrophilic property of sericin could also enhance the skin hydration capacity, which could increase the skin permeation and improve the diffusion of the anthocyanin compounds through the skin [22].

The comparison of the permeability parameters in Table 2 shows that PWCCS0.15 provided a higher permeation than the hydrogels (S3 and S4), which were indicated by the flux (0.94 ± 0.00 μg/cm²/h), the permeability coefficient (0.12 ± 0.00 cm^2^/h), and the enhancement ratio (1). Whereas S3 and S4 showed a lower permeation in all parameters except T_lag_. Therefore, S3 and S4 caused the extended permeation, since T_lag_ of S3 and S4 (1.63 ± 0.37 h and 2.98 ± 1.06 h, respectively) were higher than PWCCS0.15 (0.29 ± 0.00 h).

The permeability showed stronger evidence in the higher concentration of 0.5% PWCCS. S5 and S6 produced a higher permeability than PWCCS0.5. All parameters of PWCCS0.5 (1.07 ± 0.06 μg/cm²/h, 17.35 ± 1.72 μg/cm², 1, 0.02 ± 0.00 cm^2^/h and 2.20 ± 0.91 h) were lower than S3 and S4 (Table 2). In addition, S6 provided the highest permeability among all samples in all parameters (flux (2.12 ± 0.41 μg/cm²/h), Q24 (31.36± 2.14 μg/cm²), ER (1.98), P (0.05 ± 0.02 cm^2^/h) and T_lag_ (3.89 ± 2.76 h). Hence, these results demonstrated the improved anthocyanin permeation from the hydrogel formulations across the porcine skin. Therefore, the hydrogel containing PWCCS provided an increase in the flux and the permeability coefficient, when compared with the control or the PWCCS solutions. The higher concentration of anthocyanins in the hydrogel can significantly improve the flux and the permeability coefficient, when compared with the control. In addition, the hydrogels containing alginate (S4 and S6) presented no significant different permeability from the hydrogels without alginate (S3 and S5).

#### 2.2.3. Skin Retention of Anthocyanin

The total amount of anthocyanin (TAC) retained in the porcine skin was determined following the permeation test using the tape stripping method. The porcine skin was separated into two parts: the stratum corneum (tape) and the epidermis and dermis (skin) and the amount of anthocyanin in these parts and the amount of anthocyanin that permeated through the skin (transdermal) was determined by the pH differential method. The fractional total anthocyanin contents for formulations S2–S6 and 0.15% and 0.5% PWCCS are shown in Figure 3.

The S4 and S6 formulations that contained 0.50% PWCC, showed a higher anthocyanin permeation through the skin (transdermal TAC, 31.13 ± 1.33, 34.99 ± 1.91 µg/cm^2^, respectively) than PWCCS0.5 (transdermal TAC, 21.24 ± 2.53 µg/cm^2^). Similarly, the S3 and S4 formulations that contained 0.15% PWCC, showed a higher anthocyanin permeation through the skin (transdermal TAC, 7.54 ± 1.16 and 6.62 ± 0.22 µg/cm^2^, respectively) than PWCCS0.15 (transdermal TAC, 2.86 ± 0.37 µg/cm^2^). Therefore, the incorporation of anthocyanin in the sericin hydrogel formulations could enhance the skin permeation of anthocyanins. The analysis of the porcine skin parts indicated that anthocyanins in the solutions (PWCCS0.15 and PWCCS0.5) were preferentially retained in the stratum corneum, which is the location of keratinocytes and melanocytes. In contrast, anthocyanins in the hydrogels either permeated the skin or were retained in the epidermis and dermis, where they could affect the fibroblasts [23]. This enhanced the topical delivery of anthocyanins from the sericin hydrogels and could help maintain the anthocyanin concentrations in the application areas while prolonging the effects and reducing the frequency of administration [13].

A previous study reported that the hydrogels improved the prolonged release of anthocyanin via the anthocyanin-sericin interactions and that the swelling of the hydrogels improved the permeation of anthocyanins into the dermis through the increased water uptake. Incorporating alginate in the formulations further enhanced the anthocyanin permeation by providing a suitable viscosity and enhancing the swelling properties of the hydrogel [13].

#### 2.2.4. Skin Hydration Capacity

The skin hydration capacity of the stratum corneum area was examined in the porcine skin. The stratum corneum is the outermost layer of the epidermis and acts as a skin barrier. A comparison of untreated skin (Figure 4a) with anthocyanin treated skin (PWCCS0.15, Figure 4b), skin treated with formulations S3 (Figure 4c) and S4 (Figure 4d), revealed that the stratum corneum was thicker in the hydrogel treated samples. Therefore, it appears that the hydrogel could restore the skin barrier by hydrating the skin [24] and anthocyanin loaded into the hydrogel can readily permeate the skin [23].

### 2.3. Anti-Melanogenic Effects

#### 2.3.1. Cytotoxicity to the B16F10 Cells

The various concentrations of sericin, PWCCS, PVA, and alginate were investigated for cytotoxicity by a MTT assay on the cell viability of B16F10 cells after treatment for 24 and 48 h (Appendix A). The results indicated the concentrations which caused no cytotoxicity of sericin, PWCCS, PVA, and the alginate treatment were not more than 2%, 0.5%, 1.0% and 0.5% *w/v*, respectively. In addition, the cell viability of the B16F10 cells exposed to PWCCS0.15, PWCCS0.5, and the formulations S1–S6, was determined by a MTT assay at 24 and 48 h, and compared with the hydrogel components 2% sericin, 0.2% alginate, and kojic acid (positive control). The cell viability remained above 80% at 24 h for all treatments (Figure 5). In addition, the formulations S1–S5 increased the cell viability above 100% after treatment for 48 h, as well as PWCCS0.15 and kojic acid, except PWCCS0.5 and S6. Additionally, PWCCS0.5 became more cytotoxic after treatment for 48 h. Therefore, PWCCS0.5 was not further determined in the experiment of the melanin content and the tyrosinase inhibition.

#### 2.3.2. Tyrosinase Inhibition and the Melanin Production in the B16F10 Cells

The B16F10 cells were pretreated with 2% sericin (Ser2), PWCCS0.15, 0.2% alginate (Ag0.2), kojic acid 100 µg/mL (Kojic), and the formulations (S1–S6), for 48 h to investigate melanogenesis. Formulation S4 showed the highest tyrosinase inhibitory effect (72.22%), which was significantly more than the other tested samples and kojic acid (Figure 6a). Tyrosinase is the rate-limiting enzyme in the melanin synthesis and melanogenesis pathway [25].

Melanin, which determines the skin color in humans, is mainly produced by melanocytes located in the epidermis. Melanin is primarily synthesized in specialized organelles in melanocytes called melanosomes. Melanogenesis is a complex process that involves a series of enzymatic and chemical reactions inside melanosomes [26]. The α-melanocyte-stimulating hormone (α-MSH) is a precursor polypeptide derived from pro-opiomelanocortin that modulates pigmentation through the paracrine action, and the melanocortin 1 receptor (MC1R) is a member of the G-protein-coupled receptor family expressed on melanocytes. The binding of α-MSH to MC1R results in the activation of the adenylyl cyclase, which increases the intracellular levels of cAMP and subsequently upregulates TYR, the tyrosinase related protein-1 (TRP-1), and the tyrosinase related protein-2 (TRP-2) expression [26].

The regulation of the TRP-1 and TRP-2 expression by cAMP is directly associated with MITF, which binds to the M-box sequence (AGTCATGTGCT) located in the tyrosinase distal elements (TDEs) after its activation. Since the promoter region of MITF contains the consensus CRE sequence, the expression of MITF can also be increased by α-MSH in a cAMP-dependent manner. This demonstrates that the α-MSH-MC1R signaling pathway induces the melanin production predominantly by elevating the intracellular cAMP levels. MITF serves as the central hub of the numerous transcription factors and signaling pathways of the regulatory network of the melanin synthesis that modulates the survival, proliferation, and differentiation of melanoblasts and melanocytes [26].

Therefore, α-MSH can be used as an inducer and marker of the melanin content. The B16F10 cells treated with Ser2, PWCCS0.15, Ag0.2, S1, S2, S3, S5, and kojic acid, showed a significantly higher melanin content than the negative control, except S6. The B16F10 cells treated with α-MSH yielded the highest melanin content (135.043 ± 3.28). The melanin content of the B16F10 cells treated with S4 (101.28 ± 2.22) and S6 (93.16 ± 4.73) showed no significant difference to the negative control (Figure 6b), indicating that the hydrogel formulations consisting of 0.2% alginate and 0.15% PWCCS (S4) and 0.5% PWCCS (S6) inhibited the melanin production in the B16F10 cells. However, S4 has the most potential to inhibit the tyrosinase activity and the melanin production, as well as S6.

The sericin extracts from *Antheraea assamensis* and *Philosamia ricini* cocoons have previously been shown to inhibit tyrosinase and protect against the UV-induced damage via melanogenesis and the decreased melanin content [11]. Moreover, anthocyanin from *Hibiscus syriacus* L. has been shown to inhibit melanogenesis in the B16F10 cells via the inhibition of the tyrosinase enzymatic activity and the decreased extracellular and intracellular melanin production [19].

### 2.4. UV Protection Effects

#### 2.4.1. Cytotoxicity of the Extracts and Formulations in the HaCaT Cells

The cell viability of the HaCaT cells exposed to various concentrations of the individual formulation components (sericin, PWCCS, PVA and alginate) for 2, 6, and 24 h was assessed by a MTT assay. Sericin and PVA were found to exert no toxic effects in the concentration range 0.1–2.0%, but PWCCS and alginate caused cytotoxicity to the HaCaT cells at concentrations of more than 1.0% (Appendix A). From the results, the 6 h-treatment could be further investigated for the UV protection experiment.

The percent cell viabilities of the HaCaT cells exposed to 2% sericin (Ser2), 0.15% PWCCS (PWCCS0.15), 0.5% PWCCS (PWCCS0.5), 0.2% alginate (Ag0.2), 25 µg/mL ascorbic acid (Vit C), and the formulations S1–S6 for 6 h are shown in Figure 7. Treatment with S5 and S6 (containing 0.5% PWCCS) for 6 h, reduced the cell viability than PWCCS0.5, while S3 and S4 (containing 0.15% PWCCS) increased the cell viability more than PWCCS0.15, indicating that the concentration of PWCCS in the formulation affected the HaCaT cell viability (Figure 7). Therefore, PWCC at 0.15% concentration is appropriate for the hydrogel formulation.

#### 2.4.2. UV Protective Effects of the Extracts and Formulations on the UVB Irradiated HaCaT Cells

The effects of the UVB irradiation induced on the HaCaT cells were investigated. The MTT assay was performed by different UVB irradiations (30, 60 and 120 mJ/cm^2^). The formulations S2 and S4 were selected to investigate the UV protective effects. The cells were pretreated with 2% sericin, 0.15% PWCCS, 0.15% alginate, 25 µg/mL ascorbic acid, and the formulations (S2 and S4) on the HaCaT cells for 2, 6, and 24 h (Appendix A). The pretreatment for 6 h showed a high UV protection for all doses of the UVB-radiation (30, 60, and 120 mJ/cm^2^). Furthermore, the result of the 6 h-pretreatment revealed that the UVB irradiation dose affected the cell viability, providing UV protection. Whereas the dose of UVB irradiation at 120 mJ/cm^2^, enhanced S4 (with 0.15% PWCCS) for the cell protection after the UVB irradiation, more than S2 (without 0.15% PWCCS). In addition, PWCCS0.15 also demonstrated a strong UVB protection, as well as ascorbic acid (positive control). The cell morphology of the HaCaT cells irradiated with UVB at 30, 60, and 120 mJ/cm^2^, compared to the untreated cells are presented in Appendix A.

Figure 8 shows the cell morphology of the HaCaT cells pretreated with Vit C, sericin, 0.15% PWCCS, and the hydrogels (S2 and S4) for 6 h before the UVB irradiation at 120 mJ/cm^2,^ compared to the untreated cells and the 120 mJ/cm^2^ UVB irradiated cells. The characteristics of the cells treated with S2 and S4 are similar to the untreated cells, Ser2, Vit C, and PWCCS0.15, indicating that all of these samples provide UV protection.

The presence of sericin and PWCCS in the formulation S4 is likely to be responsible for its UV protection. According to previous research, silk sericin could inhibit the UV radiation-induced matrix metalloproteinase expression in the human dermal fibroblasts and keratinocytes [26]. Moreover, the cyanidin-3-glucoside inhibited the UVB-induction of the ROS/COX-2 pathway in the HaCaT cells and prevented the UVB-induced apoptosis [27].

### 2.5. Anti-Aging Effects

#### 2.5.1. Elastase and the Collagenase Inhibition

Elastase and collagenase are the enzymes related to skin aging and the wrinkle process, as well as the degradation of collagen and elastin. The elastase inhibition and collagenase assay were performed, as previously described with minor modifications [26]. The potential of the hydrogel formulations (S1–S6) was compared to each component in the formulation (2% sericin, 0.15% PWCCS, and 0.2% alginate) and a positive control (ascorbic acid 100 µg/mL).

All treated samples showed a significantly higher elastase inhibition than the negative control (Figure 9a). S6 showed the highest elastase inhibition (86.08 ± 2.29%). Most treated samples that contained PWCC, provided the elastase inhibition of more than 75%, as well as ascorbic acid. Therefore, PWCC might affect elastase inhibition.

For collagenase inhibition, all samples showed a lower collagenase inhibition than ascorbic acid (94.50 ± 3.26%). The hydrogel formulation containing alginate (S4 and S6) showed a higher collagenase inhibition (68.49 ± 3.38 and 65.47 ± 1.84%, respectively) than the hydrogel formulation without alginate (S3 and S5, 53.61 ± 1.91, and 50.81 ± 3.42%, respectively). The alginate in the hydrogel formulation might improve the releasing or permeation properties of anthocyanin from the hydrogels. Therefore, S4 provides the highest potential for the application in anti-elastase and anti-collagenase products (Figure 9b).

Elastase and collagenase degrade elastin and collagen to cause skin aging via the MMP-1 and MMP-2 activation. Therefore, the inhibition of MMP-1 and MMP-2 reduces skin aging through the associated inhibition of elastase and collagenase. The sericin extracts were reported to inhibit the elastase and collagenase activity by more than 50% in the fibroblast cells by down-regulating the MMP-1 expression [27].

Moreover, anthocyanins and the anthocyanin derivatives have previously shown the inhibition of elastase and the collagenase enzyme activity in the cell-free assays [28]. However, the PWCC solutions and formulations S3–S6 inhibited the elastase and collagenase activity by more than 50% in the current study. However, there is no previous report of the effect of alginate on the elastase and collagenase activity. Therefore, our research shows that the combination of sericin, PWCCS, and alginate in the formulations could improve the elastase and collagenase inhibition.

#### 2.5.2. Cytotoxicity of the Extract and the Formulations in the NHDF Cells

The cell viability of the NHDF cells exposed to various concentrations of the individual formulation components (sericin, PWCCS, PVA, and alginate) for 12, 24, and 48 h was assessed by a MTT assay. PVA and alginate were found to exert no toxic effects in the concentration range 0.1–2.0%. Whereas the suitable concentration of sericin and PWCCS depended on the incubation time (Appendix A). However, 2% sericin, 0.15% PWCSS, and 0.2% alginate are the components in the hydrogel formulations which cause no toxicity to the NHDF cells.

The percent viabilities of the NHDF cells exposed to 2% sericin (Ser2), 0.15%PWCCS (PWCCS0.15), 0.5% PWCCS (PWCCS0.5), 0.2% alginate (Ag0.2), 25 µg/mL ascorbic acid (Vit C), and the formulations S1–S6 for 12, 24, and 48 h are shown in Appendix A. All of the individual components of the hydrogel formulations (2% sericin, 0.15% PWCCS, and 0.2% alginate) and the hydrogels caused no toxicity to the NHDF cells after 12 and 24 h incubation, except the treatment with PWCCS0.5, S5, and S6 (containing 0.5% PWCCS), after 48 h incubation, that indicated the cytotoxicity on the NHDF cells. Therefore, the incubation of 24 h was further investigated for the collagen production ability.

#### 2.5.3. Collagen Content of the NHDF Cells

The 24 h treatment of 2% sericin (Ser2), 0.15% PWCCS (PWCCS0.15), 0.5% PWCCS (PWCCS0.5), 0.2% alginate (Ag0.2), 25 µg/mL ascorbic acid (Vit C), and the hydrogel formulations (S1–S6) were investigated for the collagen production on the NHDF cells by a Sircol collagen assay. The results are presented as the collagen ratio, compared with the control in Figure 10.

The treatment of the NHDF cells with the formulations S1–S6 and sericin provided significantly higher collagen ratios, more than four times that of the control. The results indicated that sericin could enhance the collagen production on the NHDF cells. Whereas PWCCS0.5 and Ag0.2 provided a lower collagen content ratio than the control. PWCCS0.5 and alginate might have no effects for the collagen production. While the lower concentration of PWCC (PWCCS0.15) resulted in the similar collagen content ratio, as well as the control (Figure 10).

However, the synergistic effects of sericin and PWCCS were obtained from S4-S6 (with PWCC), which caused a higher collagen content ratio than S1 and S2 (without PWCC). In addition, the collagen content ratio of the formulations S3 and S4 containing 0.15% PWCCS (6.06 ± 1.35 and 6.01 ± 1.41, respectively) were not significantly different from those of the formulation S5 and S6 containing 0.5% PWCCS (5.71 ± 0.09 and 5.50 ± 0.21, respectively), while S3 and S4 caused a slightly higher collagen ratio than S5 and S6. However, S3–S6 provided a significantly lower collagen content ratio than Vit C (7.52 ± 0.37).

Sericin has previously been reported to enhance the collagen type I production in fibroblasts in more than 200 µg/mL [8]. Moreover, the methionine and cysteine amino acid components of silk sericin have been shown to be important for the cell growth and collagen synthesis in the NHDF cells [29]. One study reported that the purple waxy corn ethanolic extract stimulated the production of collagen type I and enhanced the cell viability in human skin fibroblasts [5]. Anthocyanins have been shown to increase the collagen level in human skin fibroblasts and ovariectomized rats, and the immunofluorescence staining indicated that the anthocyanins stimulated the expression of the extracellular matrix (ECM) proteins, such as collagen types I and III. Cyanidin-3-glucoside (C3G) is a major compound of the anthocyanin compounds and has been reported to stimulate the production of collagen type I [30]. Collagen is the main structural component of connective tissue and the fibroblasts primarily secrete collagen type I, which is associated with skin wrinkling and aging [31].

#### 2.5.4. Evaluation of Procollagen Type I from the NHDF Cells

The quantitative evaluation of human procollagen I alpha 1 in the NHDF cells was performed with an ELISA kit. Sericin, the formulations S1–S6, and ascorbic acid induced the production of procollagen type I alpha 1, significantly more than the control untreated cells. (Figure 11). The formulations S3–S6 (with PWCC) produced procollagen type I alpha 1 contents in the NHDF cells more than the 0.15% PWCCS, 0.2% alginate, or the S1 and S2 formulations (without PWCC). The S4 formulation showed the highest procollagen type I alpha 1 content, which was significantly higher than ascorbic acid (positive control). Procollagen is a major structural protein of the skin, synthesized as a procollagen molecule in a triple helical (N and C terminal pro peptide) form. A previous report stated that a high sulfur amino acid content in sericin was associated with the increased collagen type I synthesis [29]. Moreover, a black rice extract abundant in cyanidin-3-O-β D glycoside increased procollagen type I in UV irradiated cells [32]. The combination of sericin and PWCCS in the formulations could enhance the collagen production, which was seen in the increased levels of collagen type I, and procollagen type I. Therefore, S4 was further established for the MMP-2 expression experiments.

#### 2.5.5. The Effect of the Formulation S4 on the MMP-2 Expression in the NHDF Cells

The MMP-2 expression was investigated in conditioned medium using gelatin zymography under non-reducing conditions. Sericin, PWCCS, the formulations S2 and S4, and ascorbic acid all showed a significant inhibition of MMP-2, compared to the control (Figure 12). Furthermore, the formulations S2 and S4 showed significantly higher inhibitory effects against MMP-2 than the control, PWCCS, sericin, and alginate. Moreover, S4 (75.99 ± 0.20%) showed a significantly higher inhibition than the positive control (76.61 ± 0.21%). The MMP-2 enzyme proteolytic activity in the formulations appears as a clear zone due to the lysis of the blue stained gelatin in the gel, which is compared to a MMP-2 standard, according to the molecular weight (72 KDa) [32]. Furthermore, the sericin-CMC hydrogel, as a wound dressing material, demonstrated a low level MMP-2 inhibition by zymography [30].

The sericin hydrogel with the PWCCS extract could be a new combination cosmeceutical to facilitate the skin wrinkle repair, act against skin aging, accelerate the collagen synthesis, and provide UVB-protection. The evidence showed the enhancement of the main dermal effects of the extracts (PWCCS and sericin) from previous antioxidant research [31,33]. The association of these combinations in the formulations can make a more effective therapeutic for the topical skin application with UVB-protection via the cell cycle arrest and the improved cell viability in the HaCaT cells for sunscreen products. Additionally, the sericin hydrogel with PWCCS can improve the appearance of the skin due to its anti-aging and anti-wrinkle benefits, which are mediated by the reduced MMP-2 levels, the elastase and collagenase inhibition, and the improved collagen production. Moreover, sericin and anthocyanins in the formulations were shown to inhibit the UVB irradiation effects on the HaCaT cells and stimulate collagen type I, associated with the amount of sericin from cocoons and anthocyanins in the PWCCS extracts. The formulation S4 improved the permeation and increased the collagen production in the dermal cells. In addition, this formulation reduced the melanin content via the tyrosinase inhibition of the melanogenesis pathway in the B16F10 murine melanoma cells, which also demonstrated the potential as a whitening agent.

## 3. Materials and Methods

### 3.1. Materials

3-[4,5-dimethylthiazol-2-yl]-2,5-diphenyltetrazolium bromide (MTT) and dimethyl sulfoxide (DMSO) were obtained from Thermo Scientific, (Waltham, MA, USA). The phosphate-buffered saline (PBS), fetal bovine serum (FBS), penicillin-streptomycin (10,000 U/mL), Dulbecco’s modified Eagle’s medium (DMEM), and trypsin-EDTA were obtained from Gibco (Gaithersburg, MD, USA). Sodium carbonate (Na_2_CO_3_), sodium hydrogen carbonate (NaHCO_3_), disodium hydrogen phosphate (Na_2_HPO_4_), potassium dihydrogen phosphate (KH2HPO4), and sodium chloride (NaCl) were obtained from Ajax Finechem Pty Limited (Taren Point, Australia). Polyvinyl alcohol (PVA) was obtained from Chem-Supply Pty Ltd. (Gillman, Australia). Direct red 80, picric acid, human collagen type I, ascorbic acid, L-DOPA, α-MSH, tyrosinase enzyme (≥2000 units/mg solid), collagenase from Clostridium histolyticum (0.25–1.0 FALGPA units/mg solid, ≥125 CDU/mg solid), elastase from porcine pancreas (lyophilized powder ≥ 50 units/mg protein), and gelatin type B were obtained from Sigma-Aldrich (St. Louis, MO, USA). B16F10 (murine melanoma cell line), human keratinocyte cell line (HaCaT), and primary normal human dermal fibroblasts (NHDF) were purchased from American Type Culture Collection, Manassas, VA, USA.

### 3.2. Preparation of the Extracts

The sericin and purple waxy corn cob extractions were performed, as previously described [13]. The sericin from silkworm cocoons (J108 strain) was obtained from Queen Sirikit Sericulture Center (Khon Kaen, Thailand) in January 2020. Sericin was extracted by autoclave. The cocoons were aged 42 days at collection. The silkworm cocoon was cut into small square pieces of 1 × 1 cm. Twenty-four grams of the cocoon material was added to 700 mL of purified water and autoclaved (Tomy-SX-700, Tokyo, Japan) at 120 °C, 2.5 bar for 60 min. Then, the mixture was filtered through a Whatman No. 1 filter. The filtrate was then freeze-dried at −85 °C, 0.595 bar (Labconco, Kansas, MO, USA), to obtain the sericin extract.

The purple waxy corn cobs (Thai purple waxy corn) were obtained from the Plant Breeding Research Center for Sustainable Agriculture (Khon Kaen University, Thailand) and extracted by maceration in 50% ethanol (ratio of powder to solvent 1:25), with stirring at 25 °C for 24 h. Then, the filtrate was collected. The residue was then reextracted twice following the same procedure. The filtrates were pooled, evaporated, and then freeze-dried at −85 °C, 0.595 bar (Labconco), to obtain the crude extract (purple waxy corn cob extract, PWCC) [13]. Both sericin and PWCC were stored and protected from the light at −20 °C.

### 3.3. Determination of Anthocyanins by High-Performance Liquid Chromatography with Mass Spectrometry (HPLC-MS/MS)

HPLC-MS/MS was performed using a triple quadrupole machine (API 3200 MS/MS System, ABSciex) equipped with a binary HPLC pump (Hewlett-Packard 1100, Series HPLC Value System) with software analysis, the chromatographic separation was performed on a column (Poroshell 120 SB-C18 [4.6 × 75 mm, 2.7 µm]) with a drop-in guard cartridge (Agilent Technologies, USA). The mobile phase consisted of (A) 5% formic acid and (B) methanol, were subjected to gradient elution, following: 0 to 10 min, 20% to 30% of phase B, 10 to 13 min, 30% to 100% of phase B, and 13 to 15 min, isocratic at 100% of phase B, and then a re-equilibration period of 3 min with 20% of phase B between the individual runs. The flow rate of 0.3 mL/min, a controlled column temperature of 30 °C, and an injection volume of 0.01 mL. The MS parameter conditions were followed, as previous reported [3]. The extracts were analyzed by LC-MS/MS and compared with the standard curves, such as the C3G, Pn3G, and Pg3G contents.

### 3.4. Preparation of the Sericin-Hydrogels with the Anthocyanin Extract

Six formulations (S1–S6) were prepared from stock solutions of 5% *w*/*v* sericin solution, 10% *w/v* PVA, 5% *w/v* alginate in a phosphate buffer pH 6.5, and 10% *w/v* PWCC in 50% EtOH in a phosphate buffer pH 6.5. The stock solutions were mixed to give the final concentrations of 2% *w/v* sericin, 4% *w/v* PVA, 0.00 and 0.20% *w/v* alginate, and 0.00, 0.15, and 0.50% *w/v* PWCC, according to Table 3. The final volume was adjusted to 5 mL with a phosphate buffer pH 6.5 and the mixture was stirred for 30 min at room temperature for each hydrogel formulation [13].

### 3.5. Physical Characterization of the Formulations

All formulations were evaluated for the pH, total anthocyanin content, and viscosity, according to [13].

#### 3.5.1. Swelling

The hydrogel (1 × 1 cm) was dried by vacuum drying ovens (Loboao, LDZ24T, Shanghai, China) and accurately weighed (Md). Then, they were immersed in PBS pH 7.4 at room temperature several times (0.1, 1, 2, 4, 6, 8, 12, 24, and 48 h). The dried hydrogel was removed from the plates (Mw) and the percentage swelling was calculated according to equation [34]:% Swelling = [(Mw − Md)/Md] × 100

Mw = The weight of the infiltrated hydrogels, Md = The weight of the dried hydrogels.

#### 3.5.2. In Vitro Permeation

Newborn porcine skin was obtained from the animal breeding center, Faculty of Agriculture Khon Kaen University, Thailand. The subcutaneous fat was carefully removed and the skin was cut into 3 × 3 cm^2^ squares and randomized. The skin samples were stored at −18 °C and placed at 4 °C the day before the experiments. The skin was pre-equilibrated in PBS pH 5.5 at 25 °C for 2 h before beginning the experiment [35]. The membrane permeation was performed by Franz diffusion cells with the stratum corneum side facing the donor compartment. A constant temperature (37 ± 0.5 °C) was maintained. One-gram aliquots of the samples were added into the donor chamber and PBS pH 5.5 was the sink condition in the receptor medium. A 0.7 mL sample of the receiving solution was withdrawn from the receptor chambers at predetermined intervals, i.e., 0.5, 1, 2, 4, 6, 8, 12, and 24 h. The receptor volume was replaced with an equal volume and the total anthocyanin content in the receiving solution was determined by the pH-differential method [1].

The graph between the % cumulative amount and time was constructed. The cumulative permeation steady-state flux (*J_SS_*) was obtained from the slope of the linear regression determined from the graph. The lag time (T/lag) was calculated by the linear area extrapolation to the *x*-axis intercept of the cumulative amount of the permeated profile. The other parameters, the permeability coefficient (P), including the enhancement ratio (ER), were calculated, according to the equation [35].
P = *J_SS_*/C_0_ (Initial concentration)
ER = *J_SS_* test/*J_SS_* control

#### 3.5.3. Skin Anthocyanin Retention

The amount of anthocyanin remaining in the skin was determined by the tape-stripping method. The skin was washed with PBS three times, to remove the residual donor sample. The stratum corneum (SC) was removed by attaching and removing 3M scotch tape (3M, Korea), three times [23]. The tape and skin were dissolved in 50% EtOH by sonication and the total anthocyanin was determined by the pH-differential method.

#### 3.5.4. Skin Hydration

The porcine skin was stained with haematoxylin-eosin to determine the skin hydration system of the formulations. The subcutaneous fat was removed and the formulations were applied for 24 h. Then, they were stained with haematoxylin-eosin and fixed with a 10% formalin solution and fabricated into a tissue block using a paraffin by cross section of 3–6 µm. They were observed under 200-fold magnification using a BX41TF Microscope (OLYMPUS, Tokyo, Japan) [23].

### 3.6. Anti-Melanogenic Effect

#### 3.6.1. Cytotoxicity on the B16F10 Cells by the MTT Assay

The B16F10 cells were seeded into 96-well plates at an initial concentration of 1 × 10^4^ cells/well in Dulbecco’s modified Eagle’s medium (DMEM) supplemented with 10% FBS and 1% streptomycin (100 mg/mL), and penicillin (100 U/mL), and incubated for 24 h at 37 °C and 5% CO_2_. The medium was aspirated and replaced with formulations (S1–S6), 0.15% *w*/*v* PWCC (PWCCS0.15), 0.50% *w*/*v* PWCC (PWCCS0.5), or 2.00% *w*/*v* sericin (Ser), in a culture medium (10% FBS) (1:10 ratio) to give a total volume of 100 µL. The culture medium was used as a negative control. Following 24 and 48 h of incubation at 37 °C and 5% CO_2_, the medium was removed from the 96-well plates. The cell viability was assessed by the MTT assay and the absorbance of formazan was measured at 570 nm using a microplate reader [19]. The cell viability was calculated using the following equation:% Cell viability = (As/Ac _(−) control_) × 100
where Ac is the absorbance without the extracts (control), and As is the absorbance with the extracts.

#### 3.6.2. Determination of the Melanin Content

The B16F10 cells were seeded into 6-well plates at an initial concentration of 1 × 10^5^ cells/well, in a culture medium and incubated for 24 h at 37 °C and 5% CO_2_. The formulations (S1–S5), 0.15% *w*/*v* PWCC (PWCCS0.15), 0.50% *w*/*v* PWCC (PWCCS0.5), or 2.00% *w*/*v* sericin in a culture medium (10%FBS) (1:10 ratio) to give a total volume of 1000 µL, were added with α-MSH (200 nM) then incubated for 24 h at 37 °C and 5% CO_2_. Then, the cells were washed twice with PBS and lysed with a lysis buffer (20 mM sodium phosphate (pH 6.8) and 1% Triton X-100) and then centrifuged at 12,000 rpm for 15 min. The pellets were dissolved in 1 N NaOH containing 10% DMSO for 1 h at 80 °C, to dissolve the melanin. The protein content in the supernatant was determined by the BCA method, the melanin content was normalized to the amount of protein in same reaction [21].

#### 3.6.3. Tyrosinase Activity

The B16F10 cells (1 × 10^5^ cells/well) were incubated in 6-well plates for 24 h. The samples were added and incubated at 37 °C and 5% CO_2_ for 24 h. The cells were washed twice with PBS and lysed in 200 µL with 0.1 M PBS pH 6.8 containing 0.1% tritoxX-100. Following the cell lysis, the protein content was determined by the BCA method and 100 µg/mL was mixed with 100 µL of 0.1% L-DOPA in PBS pH 6.8, and incubated for 20 min at 37 °C. The absorbance was measured at 475 nm with a microplate reader and calculated as the tyrosinase inhibition [15].

### 3.7. UV Protection Effects

#### 3.7.1. The Cytotoxicity on the HaCaT Cells by the MTT Assay

The cell viability of the formulations was investigated on the HaCaT cells. The cells were seeded into a 96-well plate at an initial concentration of 1.5 × 10^4^ cells/well in DMEM supplemented with 10% FBS and 1% streptomycin (100 mg/mL), and penicillin (100 U/mL), and incubated at 37 °C and 5% CO_2_ for 24 h. The medium was aspirated and replaced with a sample. The formulations (S1–S6), 0.15% *w/v* PWCC (PWCCS0.15), 0.50% *w*/*v* PWCC (PWCCS0.5), or 2% *w*/*v* sericin in a culture medium (1:10 ratio), to give a total volume of 100 µL. The culture medium was used as the negative control ((−) control). Then, the cells were incubated at 37 °C and 5% CO_2_ for 24 h. The medium was removed. The MTT was added and the formazan absorbance was measured at 570 nm using a microplate reader [12]. The cell viability was calculated using the following equation:% Cell viability = (As/Ac) × 100(1)
where Ac is the absorbance of media without the extracts (control), and As is the absorbance with the extracts.

#### 3.7.2. Effects of the UVB Irradiation on the HaCaT Cells

The HaCaT cells were seeded into 6-well plates at a density of 1.5 × 10^5^ cells/well. The medium was then removed and the cells were covered with 0.5 mL of PBS. Then, the cells were exposed to a UVB lamp (Philip UVB Narrowband 311 nm phototherapy lamp) at 311 nm for 10 min. The irradiance was measured using a photoradiometer (UV 340B) [20]. For the prevention of overheating during the irradiation, the microplate was kept on ice.

#### 3.7.3. UV Protection Effects on the HaCaT Cells

The protection of the formulations against the UVB-induced cytotoxicity on the HaCaT cells was performed with the MTT assays. The HaCaT cells were treated with the formulations (S1–S5), 0.15% *w/v* PWCC (PWCCS0.15), 0.50% *w*/*v* PWCC (PWCCS0.5), or 2.00% *w/v* sericin (Ser), in a culture medium for 2, 6, and 24 h at 37 °C with 5% CO_2_. The medium was then removed and covered with 0.5 mL of PBS. Then, the cells were exposed to a dose, at 120 mJ/cm^2^, of UVB radiation, for 10 min. Then, the cells were incubated for 24 h and determined for the cell viability.

### 3.8. Anti-Aging Effects

#### 3.8.1. Elastase Inhibition Assay

The elastase inhibition assay was performed, as previously reported, with modifications [27]. Briefly, the porcine pancreatic elastase (4.80 U/mg) was purchased from Sigma-Aldrich and 2.4 U/mg of the enzyme stock was prepared in a tris-HCL buffer pH 8.50 and incubated with the samples. A stock solution of 8 mM of N-succinyl-(Ala) 3 -p nitroanilide was prepared in the same buffer. The final concentration of 0.2 U/mg of the enzyme in 100 µL and 10X diluted of the hydrogel, 50 µL of the extracts were added to each well and then incubated for 30 min, the 50 µL of 2 mM substrate were added to the mixture and then incubated for 30 min, and the absorbance was measured at 410 nm using a microplate reader. The elastase activity was calculated using the following equation:Elastase inhibition (%) = {[(A − B) ± (C − D)]/(A − B)} × 100
where A is the absorbance without the extracts (control), B is the absorbance without the extracts and enzyme (blank of A), C is the absorbance with the extracts, and D is the absorbance with the extracts and without the enzyme.

#### 3.8.2. Collagenase Inhibition Assay

The in vitro collagenase inhibition assay was performed by a modified protocol of Kumar and Mandal [27]. Briefly, collagenase (E.C. 3.4.24.3) from Clostridium (Type IA) specific activity 1U/mg was purchased from Sigma-Aldrich. One U of the enzyme stock in a tricine buffer (0.05M pH 7.5 containing 0.4 M NaCl and 0.01M CaCl_2_) was incubated with the samples. The stock 0.8 mM N-[3-(2-Furyl) acryloyl]-Leu-Gly-Pro-Ala (FALGPA) solution was prepared in the same buffer. The final concentration of 0.2 U of the enzyme 25 µL, 10X diluted of hydrogel 25 µL and buffer 25 µL, were added to each well and then incubated for 15 min, 50 µL of the 2 mM substrate were added to the mixture to immediately measure the absorbance at 340 nm, using a microplate reader. The collagenase activity was calculated using the following equation:Collagenase activity (%) = 100 − [{(As − Ab)/(Ac − Ab)} × 100]
where As is the absorbance of the test sample, Ab is the absorbance of the blank (without the substrate), and Ac is the absorbance of the substrate without collagenase.

#### 3.8.3. The Cytotoxicity on the NHDF Cells by the MTT Assay

The cells were seeded into a 96-well plate at an initial concentration of 1.5 × 10^4^ cells/well in DMEM supplemented with 10% FBS and 1% streptomycin (100 mg/mL), and penicillin (100 U/mL), and incubated at 37 °C and 5% CO_2_ for 24 h. The medium was aspirated and replaced by a sample. The formulations (S1–S6), 0.15% *w/v* PWCC (PWCCS0.15), 0.50% *w/v* PWCC (PWCCS0.5), or 2% *w/v* sericin in a culture medium (1:10 ratio) to give a total volume of 100 µL. The culture medium was used as the negative control ((−) control). Then, the cells were incubated at 37 °C and 5% CO_2_ for 24 h. The medium was removed. The MTT was added and the formazan absorbance was measured at 570 nm using a microplate reader [12]. The cell viability was calculated using the following equation:% Cell viability = (As/Ac) × 100
where Ac is the absorbance of media without the extracts (control), and As is the absorbance with the extracts.

#### 3.8.4. Collagen Production in the NHDF Cells

The NHDF cells were seeded into 96-well plates at a density 1.5 × 10^4^ cells/well in a culture medium and incubated for 24 h at 37 °C and 5% CO_2_. The cells were treated with the test samples and incubated for 24 h at 37 °C. Ascorbic acid was used as a positive control. Then, 24 h later, the collagen content of the supernatant from the culture medium was determined by a Sircol collagen assay. Briefly, the supernatant was dissolved in 0.1 M acetic acid (1:1) and Sirius red was reacted with the collagen. Then, the mixture was centrifuged at 12,000 rpm for 5 min and washed with 0.1 M HCl. Then, it was centrifuged at 12,000 rpm for 5 min and dissolved in 0.5 N NaOH. The absorbance was measured using a microplate reader at a wavelength of 550 nm. All experiments were performed in triplicate. The amount of collagen was calculated, based on a standard curve of the soluble collagen: standard bovine collagen type I [9].

#### 3.8.5. Measurement of the Human Procollagen Alpha I

The human procollagen alpha I levels were determined in the NHDF cells in a cell culture medium using an enzyme-linked immunosorbent assay (ELISA), according to the manufacturer’s protocols (Abcam, Biomed Diagnostics, Cambridge, MA, USA).

#### 3.8.6. MMP-2 Expression on the Zymography Technique

The gelatinase (MMP-2) activity was determined by gelatin zymography. The cells were cultured in a DMEM medium to 80% confluency. Then, the culture supernatant was collected. The culture supernatant was used to determine the gelatinase activity. Then, the loading buffer containing 10% SDS under a non-reducing condition was added and loaded on 7.5% gelatin zymography. Then, 2.5% triton X-100 will be used for the gel washing with the activating buffer (50 mM Tris-HCL; pH 7.5, 200 mM NaCl and 5 mM CaCl_2_). The gels were stained with Coomassie brilliant blue R-250 [27].

### 3.9. Data Analysis

The data were analyzed by SPSS version 28 (SPSS Inc., Chicago, IL, USA; licensed KKU software) and are presented as the mean ± standard error of mean (SEM). The differences between the groups were assessed by a one-way analysis of variance (ANOVA) followed by Duncan’s post hoc test. The comparison in the stability test was evaluated by a paired sample *t*-test. A *p*-value of less than 0.05 was considered statistically significant.

## 4. Conclusions

A summary of the effects of the hydrogel formulation (S4) containing sericin with the PWCCS extract is shown in Figure 13. The S4 formulation showed an enhanced permeation into the skin, which sealed and improved the surface area of the skin, promoted the hydration capacity, and the improved anthocyanin was associated with the collagen production in the NHDF cells through the down-regulation of MMP-2, without cytotoxicity. In addition, S4 showed a high degree of collagenase and elastase inhibition, which supports its application as an anti-wrinkle and anti-aging product. The whitening effects of S4 were confirmed via the tyrosinase inhibition, and a reduction in the melanin pigment content without cytotoxicity in the B16F10 cells. These effects included the UVB-protection, which supports its application as a sunscreen product. S4 also increased the cell viability and restored the cell cycle arrest in the HaCaT cells. These results represent a promising basis for clinical trials in the future.

## Figures and Tables

**Figure 1 pharmaceuticals-16-00035-f001:**
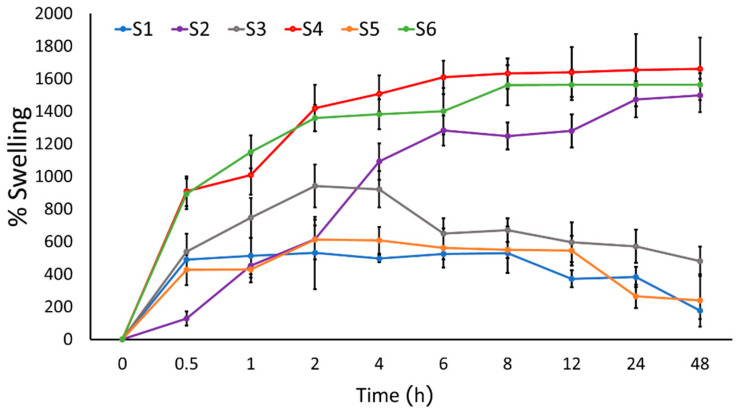
Swelling profiles of the sericin hydrogel formulations S1–S6 in PBS pH 7.4 (*n* = 3).

**Figure 2 pharmaceuticals-16-00035-f002:**
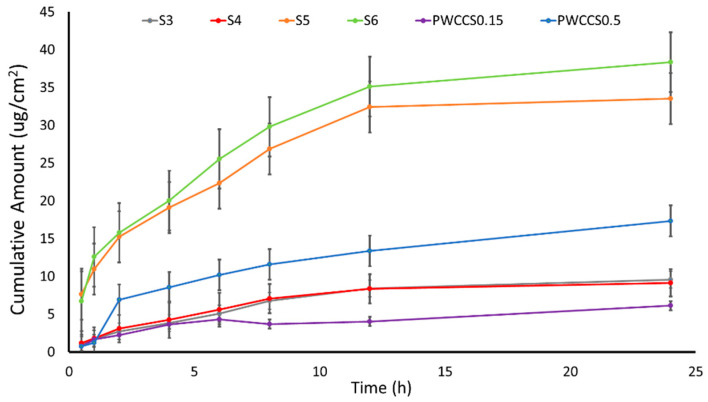
The anthocyanin permeation profiles for PWCCS0.15, PWCCS0.5, and the sericin hydrogels S3–S6 through the porcine skin in PBS pH 5.5 (*n* = 3).

**Figure 3 pharmaceuticals-16-00035-f003:**
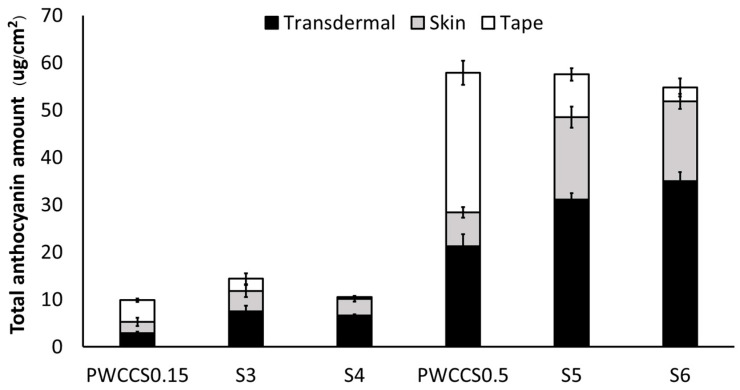
Amount of anthocyanin in the stratum corneum (tape), epidermis and dermis without the stratum corneum (skin), and permeated through the skin (transdermal) of porcine skin treated with 0.15% PWCCS (PWCCS0.15), 0.5% PWCCS (PWCCS0.5), or sericin hydrogels S3–S6, in PBS pH 5.5.

**Figure 4 pharmaceuticals-16-00035-f004:**
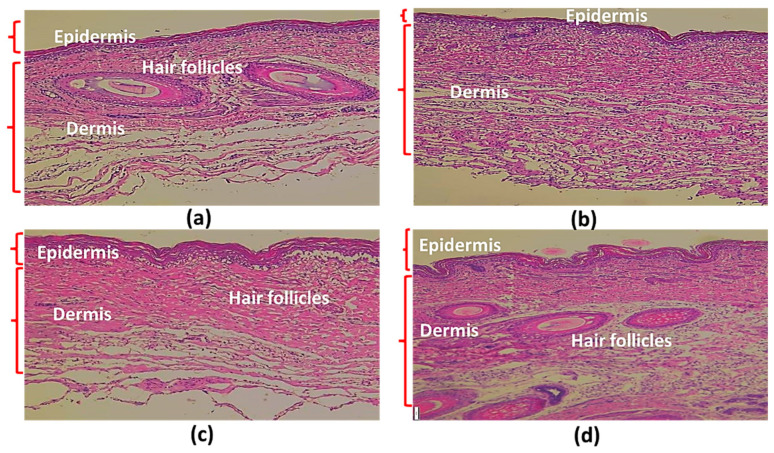
Sections of porcine skin. Untreated skin (**a**), skin treated with PWCCS0.15 (**b**), skin treated with S3 (**c**) and with S4 (**d**).

**Figure 5 pharmaceuticals-16-00035-f005:**
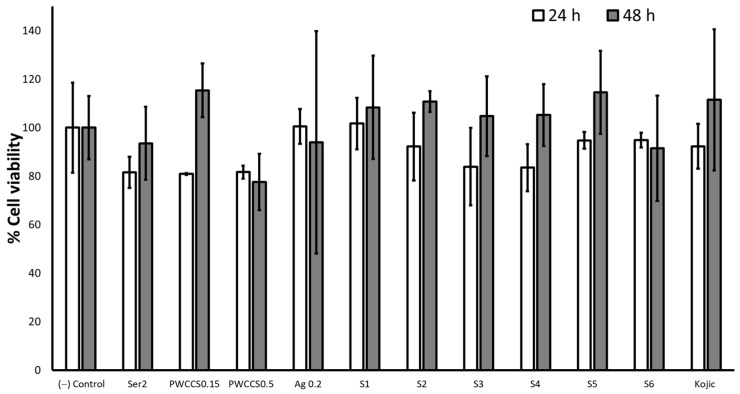
Effect of 2% sericin (Ser2), 0.15% PWCCS (PWCCS0.15), 0.5% PWCCS (PWCCS0.5), 0.2% alginate (Ag0.2), 100 µg/mL kojic acid (Kojic), and the hydrogels S1–S6 on the cell viability of the B16F10 cells after 24 and 48 h. Data represent the mean ± SEM of the three replicates.

**Figure 6 pharmaceuticals-16-00035-f006:**
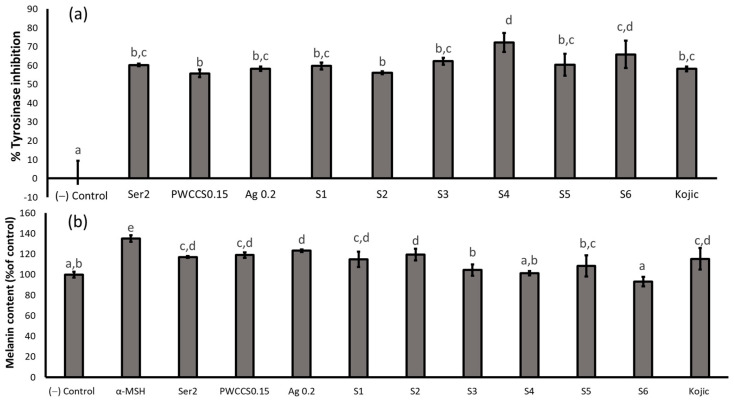
Effect of the α-melanocyte-stimulating hormone (α-MSH), 2% sericin (Ser2), 0.15% PWCCS (PWCCS 0.15), 0.2% alginate (Ag0.2), 100 µg/mL kojic acid (Kojic), and the hydrogels S1–S6 treatment for 48 h on the (**a**) tyrosinase activity (% inhibition) and (**b**) melanin content on the B16F10 melanoma cells. Data represent the mean ± SEM of the three replicates. Statistical significance was evaluated by Duncan’s post-hoc test; a–e letters indicate significant differences between the group at *p* < 0.05.

**Figure 7 pharmaceuticals-16-00035-f007:**
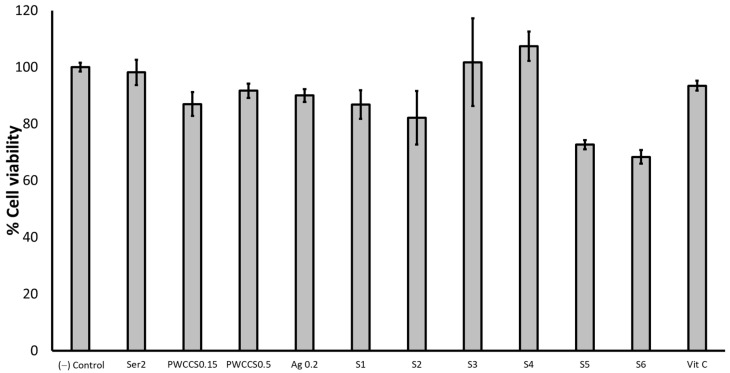
Viability of the HaCaT cells exposed to 2% sericin (Ser2), 0.15% PWCCS (PWCCS0.15), 0.5% PWCCS(PWCCS0.5), 0.2% alginate (Ag0.2), 25 µg/mL ascorbic acid (Vit C), and the formulations S1–S6 for 6 h. Data represent the mean ± SEM of the three replicates.

**Figure 8 pharmaceuticals-16-00035-f008:**
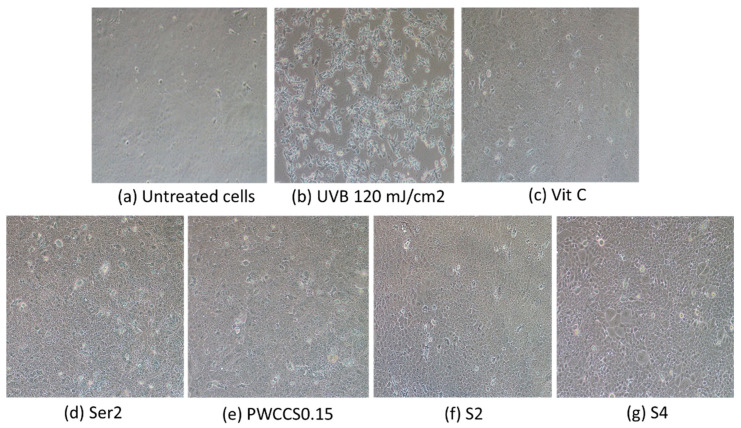
Morphology of the HaCaT cells pretreated with the hydrogels and components for 6 h before the UVB irradiation at 120 mJ/cm^2^: (**a**) untreated cells, (**b**) cells irradiated with 120 mJ/cm^2^, (**c**) cells treated with 25 µg/mL ascorbic acid, (**d**) cells treated with 2% sericin (Ser2), (**e**) cells treated with 0.15% PWCCS (PWCCS0.15), (**f**) cells treated with S2, (**g**) cells treated with S4.

**Figure 9 pharmaceuticals-16-00035-f009:**
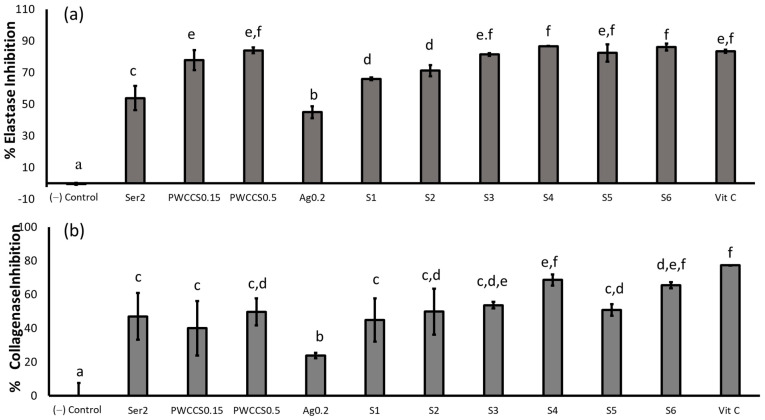
Elastase (**a**) and collagenase (**b**) inhibitory effects of the hydrogel formulations and their components in the B16F10 melanoma cells treated with 2%sericin (Ser2), 0.15%PWCCS (PWCCS0.15), 0.5% PWCCS (PWCCS0.5), 0.2% alginate (Ag0.2), the formulations S1–S6, or ascorbic acid 100 µg/mL (Vit C). Data represent the mean ± SEM of the three replicates. Statistical significance was evaluated by Duncan’s post-hoc test; a–f letters indicate significant differences between the group at *p* < 0.05.

**Figure 10 pharmaceuticals-16-00035-f010:**
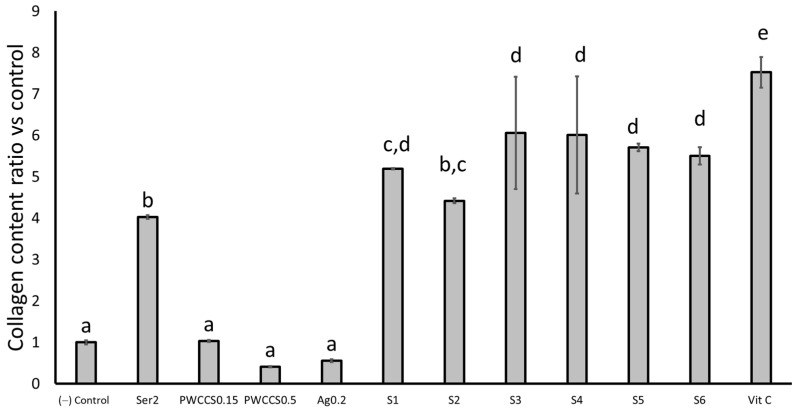
Collagen content ratio vs. the control of the NHDF cells after 24 h treatment with 2% sericin (Ser2), 0.15% PWCCS (PWCCS0.15), 0.5% PWCCS (PWCCS0.5), 0.2% alginate (Ag0.2), 25 µg/mL ascorbic acid (Vit C), and the formulations S1–S6. Data represent the mean ± SEM of the three replicates. Statistical significance was evaluated by Duncan’s post-hoc test; a–e letters indicate significant differences between the group at *p* < 0.05.

**Figure 11 pharmaceuticals-16-00035-f011:**
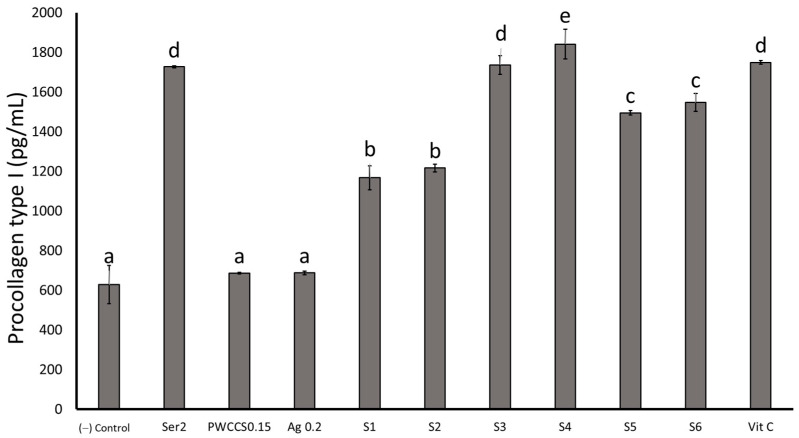
Procollagen type I content of 2% sericin (Ser2), 0.15% PWCCS(PWCCS0.15), 0.2% alginate (Ag0.2), ascorbic acid 25 µg/mL (Vit C), S2, and S4 on the NHDF cells after 24 h. Data represent the mean ± SEM of the three replicates. Statistical significance was evaluated by Duncan’s post-hoc test; a–e letters indicate significant differences between the group at *p* < 0.05.

**Figure 12 pharmaceuticals-16-00035-f012:**
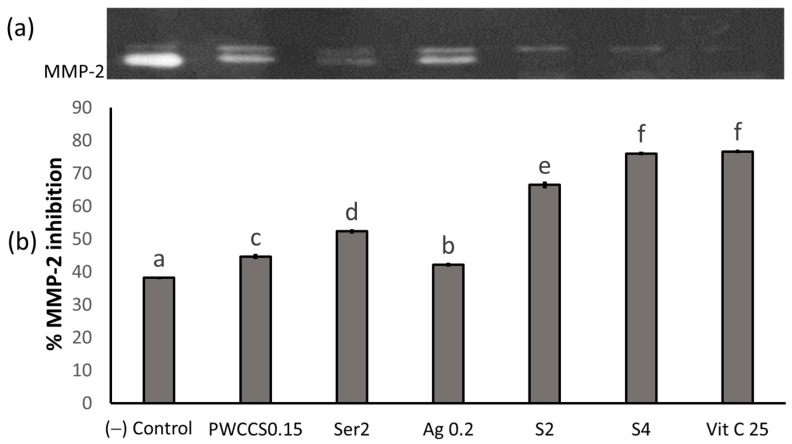
MMP-2 expression by the zymography detection content of 0.15% PWCCS(PWCCS0.15), 2% sericin (Ser2), 0.2% alginate (Ag0.2), 25 µg/mL ascorbic acid (Vit C), S2, and S4 treated on the NHDF cells after 24 h. (**a**) MMP-2 zymography bands; (**b**) MMP-2 inhibition analyzed the band intensity by the image J analysis. Data represent the mean ± SEM of the three replicates. Statistical significance was evaluated by Duncan’s post-hoc test; a–f letters indicate significant differences between the group at *p* < 0.05.

**Figure 13 pharmaceuticals-16-00035-f013:**
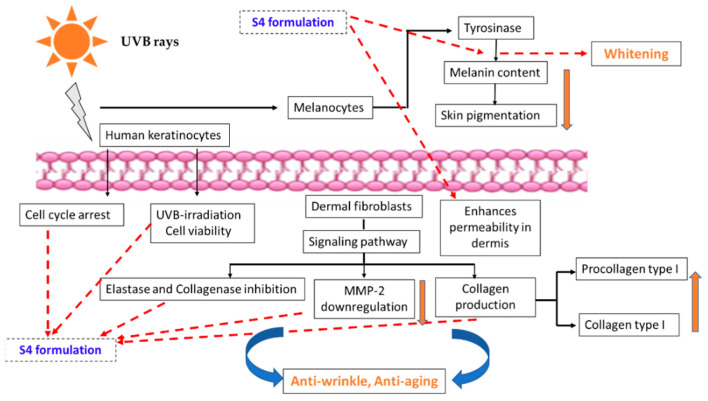
Summary of the main pathways of the UVB-induced effects on human keratinocytes, dermal fibroblasts, and melanocytes, and the potential mechanisms of the S4 hydrogel whitening and anti-aging effects.

**Table 1 pharmaceuticals-16-00035-t001:** Physical characteristics of formulations S1–S6.

Formulation	Conc of PWCCS (%)	pH	Viscosity (Pa.S)	Total Anthocyanin (mg C3GE/L)
S1	-	6.55 ± 0.00	35.52 ± 0.00 ^e^	-
S2	-	6.62 ± 0.10	37.51 ± 0.01 ^f^	-
S3	0.15	6.57 ± 0.02	35.39 ± 0.02 ^c^	41.48 ± 6.62 ^a^
S4	0.15	6.71 ± 0.28	35.50 ± 0.03 ^d^	39.21 ± 3.26 ^a^
S5	0.50	6.47 ± 0.00	34.27 ± 0.04 ^a^	107.61 ± 11.98 ^b^
S6	0.50	6.45 ± 0.05	34.49 ± 0.05 ^b^	102.87 ± 1.53 ^b^

These data represent the mean ± SEM of the three replicates. ^a–f^ letters indicate significant differences in the same column at *p* < 0.05 (by a one-way ANOVA).

**Table 2 pharmaceuticals-16-00035-t002:** Permeability parameters of PWCCS0.15, PWCCS0.5, and the hydrogel formulations S3, S4, S5, and S6.

Formulation	Flux (μg/cm²/h)	*Q*24 (μg/cm²)	ER	P (cm^2^/h)	T_lag/h_
PWCCS0.15	0.94 ± 0.00 ^b^	7.02 ± 0.00 ^a^	1.00	0.12 ± 0.00 ^b^	0.29 ± 0.00 ^a^
S3	0.52 ± 0.32 ^a^	7.73 ± 2.65 ^a^	0.55	0.04 ± 0.03 ^a^	1.63 ± 0.37 ^b^
S4	0.57 ± 0.06 ^a^	6.98 ± 0.64 ^a^	0.60	0.07 ± 0.01 ^a^	2.98 ± 1.06 ^c^
PWCCS0.5	1.07 ± 0.06 ^a’^	17.35 ± 1.72 ^a’^	1.00	0.02 ± 0.00 ^a’^	2.20 ± 0.91 ^a’^
S5	2.10 ± 0.09 ^b’^	28.54 ± 6.98 ^b’^	1.96	0.04 ± 0.00 ^b’^	3.57 ± 2.76 ^a’^
S6	2.12 ± 0.41 ^b’^	31.36 ± 2.14 ^b’^	1.98	0.05 ± 0.02 ^b’^	3.89 ± 2.76 ^a’^

*Q*24: cumulative amount permeated at 24 h; ER: enhancement ratio; P: permeability coefficient; T_lag_: lag time; PWCCS0.15: 0.15% purple waxy corn cob extract solution; PWCSS0.5: 0.5% purple waxy corn cob extract solution; S3–S6: hydrogel formulations. ^a, b, c, a’, b’^ letters indicate the significant differences between the group at *p* < 0.05 (by a one-way analysis of variance (ANOVA) followed by Duncan’s post hoc test). All values are mean ± SEM (*n* = 3).

**Table 3 pharmaceuticals-16-00035-t003:** The composition of the sericin hydrogel formulations.

Formulation	5% *w*/*v* Sericin (mL)	10% *w*/*v* PVA (mL)	5% *w*/*v* Alginate (mL)	10% *w*/*v* PWCCS (mL)	Final Volume (mL)	Final Conc. of Alginate (%)	Final Conc. of PWCCS (%)
S1	2.00	2.00	0.00	0.000	5.00	0.00	0.00
S2	2.00	2.00	0.20	0.000	5.00	0.20	0.00
S3	2.00	2.00	0.00	0.075	5.00	0.00	0.15
S4	2.00	2.00	0.20	0.075	5.00	0.20	0.15
S5	2.00	2.00	0.00	0.250	5.00	0.00	0.50
S6	2.00	2.00	0.20	0.250	5.00	0.20	0.50

## Data Availability

Data is contained within the article and Appendix A.

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
