# Peer review of "The Potential of Purple Waxy Corn Cob (Zea mays L.) Extract Loaded-Sericin Hydrogel for Anti-Hyperpigmentation, UV Protection and Anti-Aging Properties as Topical Product Applications"

_pharmaceuticals, 2022, doi:10.3390/ph16010035_

Round 1
Reviewer 1 Report
The research work entitled “The potential of purple waxy corn cob (Zea mays L.) extract 2 loaded-sericin hydrogel for anti-hyperpigmentation, UV protection and anti-aging properties as topical product applications” aimed to develop purple waxy corn (Zea mays L.) cob extract loaded sericin has been well designed and executed. Ample tests/methodology has been applied for the evaluation of this topical preparation. The results have supported the potential use of the product for anti-hyperpigmentation, UV protection and anti-aging effects. Therefore, I would recommend the manuscripts for publication.
Author Response
Thank you so much for your kind review.
Reviewer 2 Report
In this work, the properties of six formulations of sericin-based hydrogels and corn anthocyanin extracts were analyzed to evaluate the anti-melanogenesis action, for the UV protection of human keratinocytes and to analyze the anti-aging activity on human dermal fibroblasts. . These preparations have been formulated as cosmetic products. The authors intend to demonstrate the synergistic effects of anthocyanin derivatives (antioxidants, anti-inflammatories), with the carrier properties of sericin-based hydrogels, whose inhibition of hyaluronidase and elastase appear already demonstrated. The work appears self-consistent and reaches a sufficient scientific level to be accepted. I suggest two minor changes:
i) the first to report the molecular structures of the indicated anthocyanin derivatives.
ii) Modify the sentence: "a purple corn silk extract", which appears unsuitable to properly define the extract to which the authors refer.
Author Response
Please see in the attached file.
